# Why Do Key Decision-Makers Fail to Foresee Extreme 'Black Swan' Events? A Case Study of the Pike River Mine Disaster, New Zealand

**Richard John Logan [1,\*], Robert Y. Cavana [1], Bronwyn E. Howell [1]** and **Ian Yeoman [2]**

1   School of Management, Victoria University of Wellington, Wellington P.O. Box 600, New Zealand;
    bob.cavana@vuw.ac.nz (R.Y.C.); bronwyn.howell@vuw.ac.nz (B.E.H.)
2   Hospitality and Tourism, NHL Stenden University of Applied Sciences, 8900 CB Leeuwarden,
    The Netherlands; ian.yeoman@nhlstenden.com
\*   Correspondence: richard.logan@vuw.ac.nz

**Abstract:** This research addresses the strategic issue of why key decision-makers fail to foresee potential extreme 'black swan' events. Following a review of the literature, a conceptual framework is developed that identifies two types of organisational blindness that are reflected in Tetlock's hedgehog cognitive thinking style, being the oversimplification of uncertainty (e.g., inductive biases) and an unquestioned, top-down, reference narrative. This framework is tested using a case study approach and qualitative analysis of secondary data sources available from the Royal Commission of Inquiry and other published reports following the 2010 methane explosion at the Pike River Coal Ltd.'s mine (Pike) in New Zealand, that killed 29 miners and caused the loss of all funds invested. The results indicate that the combined effect of both blindnesses meant that Pike's collective intelligence was limited, and for the three key decision-makers at the Pike River mine, some type of extreme 'black swan' event was apparently inevitable. This research provides theoretical and practical contributions to the analysis of business and public policy decision-making under uncertainty.

**Keywords:** Knightian uncertainty; uncertainty aversion; fox/hedgehog cognitive thinking styles; narrative economics; complex systems; decision-making under uncertainty; strategic drift; black swan events; coal mine disaster

## 1. Introduction

This research addresses the question of why key decision-makers fail to foresee, listen to the warnings of or act upon a surprising extreme event, or 'black swan' event. One possible answer lies in the various forms of organisational bounded rationality under high uncertainty. Where there is strategic drift [1], where reality has drifted away from an organisation's vision and strategy, there is increased potential for one or several 'black swan' events. Hence the question is, why do otherwise good boards, management teams and key decision-makers get into the cognitive blindness of strategic drift in the first place?

A conceptual framework was developed to address this question, and a case study was used to test the framework. The case study considered was the 2010 methane explosion at the Pike River Coal Ltd.'s mine (Pike), near Greymouth, New Zealand [2], that killed 29 miners, causing the loss of all funds invested (over $300 m). Secondary data available from the published reports of the Royal Commission of Inquiry and other published sources based on interviews and company records was analysed qualitatively, using a 'bricoleur' approach. The study identified two highly interconnected types of organisational blindness: unquestioned inductive cognitive biases (which affected 'how' the key decision-makers thought); and a single, unquestioned, top-down, reference narrative (which affected 'what' the key decision-makers thought). Both 'how' and 'what' simplifications can overlook key risks and uncertainties. The combined effect of both blindnesses meant that for the

three key decision-makers at Pike, being the Board Chair, the Chief Executive and General Manager (Mines), some type of negative 'black swan' event occurring appeared inevitable. This research suggests that Pike's three key decision-makers had adopted an overconfident, 'hedgehog' cognitive thinking style (as per Tetlock [3]) which limited Pike's collective intelligence to the small in-group, rather than encouraging wider input at a time of great situational complexity and uncertainty.

The paper proceeds as follows. The next section contains a literature review and outlines the development of the conceptual framework for this research. This is followed by a brief overview of the case-study, being the decision-making around the Pike River coal mine disaster in 2010, which involved a subsequent Royal Commission of Inquiry [4]. An overview of the qualitative research methods is then outlined. The next section outlines the main findings from the research study and some concluding comments are provided in the final section.

## 2. Conceptual Framework Development

This section considers organisational decision-making under uncertainty where there is a natural tendency to simplify the complexity and uncertainty. It does this by using the Knightian distinction of risk versus uncertainty, overlaid with uncertainty aversion, which is the tendency to default to the known, i.e., away from uncertainty to risk.

### 2.1. The Two Fundamentally Different Approaches to Knightian Risk/Uncertainty

Various authors (e.g., Bridge [5], Knight [6], Kay and King [7]) describe two fundamentally different schools of thought and/or approaches to business uncertainty, involving not only different concepts and perspectives but essentially different philosophies.

- Knightian risk—resolvable uncertainty. The focus is on the known and the expected. This is ideal for predictable environments, where uncertainty is reduced by mathematical quantification of risk, modelling, and prediction. Models specifying risk tend to dominate the decision-making literature.
- Knightian uncertainty—radical uncertainty. The focus is on the unknown and the unexpected. This is ideal for unpredictable environments. Radical uncertainty is where the outcome is uncertain and the degree of risk cannot be assessed. This approach accepts the uncertainty, and is prepared to react to reduce any harm from the unexpected, but also to maximise its possible advantage. It uses ideas such as exploration, antifragility and 'trial and error'.

Currently, there is a paradox. On the one hand, Hodgson [8] (p. 170) comments on the decades-long decline of the Knight–Keynes concept of uncertainty in mainstream economics due to the overwhelming dominance of modelling which needs the quantification of risk. On the other hand, the importance of radical uncertainty can be seen in unexpected events, such as rare 'black swan' events [9], as well as frequent 'unscheduled or novel' events. Mangee [10] (p. 8) found that unscheduled novel events comprised over two thirds of events explicitly identified in Dow Jones news reports influencing US corporate prospects and over four-fifths of macro events for the US were considered unscheduled and novel. Mangee [10] believes this drives the stock market.

Bridge [5] believes

> 'that adopting a Knightian uncertainty approach requires an acknowledgement that uncertainty is the norm, not the exception. Therefore, it is not like switching from one technique to another, but instead requires adoption of a different way of seeing things: a conscious abandonment of a constant search for certainty and instead an explorer's acceptance of the possibilities in uncertainty'.

A second reason (and maybe the most important reason) why decision-makers instinctively favour Knightian risk approaches is the 'uncertainty aversion' bias.

### 2.2. The 'Uncertainty Aversion' Bias Causes a Default from Radical to Resolvable Uncertainty

'Uncertainty or ambiguity aversion' is the decision-makers' tendency to favour the known uncertainties over the unknown ones, including known risks which are able to have probabilities assigned [11]. This is a deep and instinctive aversion. Uncertainty aversion is reflected in proverbs such as 'better the devil you know than the devil you don't'. Uncertainty aversion is one of the most striking decision biases in which people show a systematic tendency to avoid options for which the level of risk is unknown [12–15]. Gigerenzer [16] believes that uncertainty aversion creates an illusion of greater certainty than there actually is. The focus is on what is known and certain. What is unknown or uncertain is minimised or abstracted away, which could be a problem if that leads to a conscious or unconscious reduction in the level of uncertainty acknowledged.

### 2.3. Decision-Makers Have Either a High (i.e., Fox) or Low (i.e., Hedgehog) Tolerance to Uncertainty

Pulford and Colman [12] believe that uncertainty tends to induce a disturbing and aversive psychological state in decision-making. However, there are large individual differences in intolerance of uncertainty. Tetlock [3] suggests that in dealing with uncertainty there are two distinct cognitive thinking styles based on the decision-maker's preferred states of knowing. He calls these fox/hedgehog cognitive thinking styles. The fox/hedgehog categorisation is based on a fragment of a Greek poem by Archilochus (c.660 BC). This research puts emphasis on the fox/hedgehog attitude to uncertainty.

Tetlock [3] describes decision-makers as hedgehogs if they prioritise the known and certain and they avoid uncertainty. Hedgehogs dislike dissonance and prefer to organise the world into neat evaluative gestalts. Hedgehogs have a strong need for structure and closure and are most likely to rely on their preconceptions when interpreting new situations since they hold strong relevant attitudes (i.e., prior biases) [3,17] (pp. 32, 82). When operating under ambiguity, hedgehogs find it hard to resist filling in the missing data points with ideological scripted event sequences [3,18] (p. 38). This means they tend to reduce all challenges and dilemmas to simple top-down deductive narratives which they seek to defend. For a hedgehog, anything that does not fit with their organising idea is discarded. This makes hedgehog cognitive decision-makers especially prone to surprise events.

Tetlock [3] describes decision-makers as foxes when they deal with complexity by drawing from an eclectic array of traditions and approaches. Consequently, foxes are better equipped to survive in rapidly changing environments because they can quickly abandon bad or flawed ideas. Hedgehogs prefer to be masters of the known, so they can become very confident with the correctness of their judgements. Foxes however have greater awareness of what they do not know, so they are usually modest and less bold in their viewpoints [3].

This research found there were two main ways 'uncertainty aversion' can be detected in organisations: first, unquestioned inductive thinking/inductive cognitive biases (affecting 'how' key decision-makers thought), and second, an unquestioned reference narrative (affecting 'what' key decision-makers thought).

### 2.4. Unquestioned Inductive Cognitive Biases Can Cause a Default from Radical to Resolvable Uncertainty

Whilst there are numerous ways uncertainty is simplified, this study focussed on the psychological shortcuts, especially those involving induction and inductive biases such as optimism bias, confirmation bias, sunk cost fallacy, and the planning fallacy [19]. Decision-makers may unconsciously be affected by the 'problem of induction' as identified by the philosopher Hume [20]. This is where decision-makers only consider in-sample data and this makes them prone to the surprise of out-of-sample information/events, i.e., 'black swan' events [9].

### 2.5. Unquestioned Top-Down Reference Narratives Can Cause a Default from Radical to Resolvable Uncertainty

Like business or economic models, narratives inherently simplify complexity and uncertainty [21]. Mangee [10] (p. 3) believes that 'Where there is uncertainty there are narratives—narratives are the currency of uncertainty'. One aspect of this is what Kay and King [7] call an organisation's 'reference narrative', being the collective management/board mindset. The reference narrative is not constructed in isolation, but will be discussed with friends and colleagues, and will be subject to professional advice and the collective intelligence accumulated and available in the various communities in which the decision-makers live. People change their reference narrative in response to disconfirming events, but infrequently and discontinuously [7]. Under conditions of certainty, there may not be any pressure to update the reference narrative; but under conditions of uncertainty, it is essential to be aware of the need to revise and update the reference narrative as new information becomes available. Hedgehog cognitive decision-makers, who come with 'strong prior biases' (i.e., an unchallenged reference narrative), are especially surprised by the unexpected [3] (pp. 179, 222). The two factors leading hedgehog cognitive thinkers to default from radical uncertainty to resolvable uncertainty and therefore making them blind to 'black swan' events are shown diagrammatically in Figure 1.

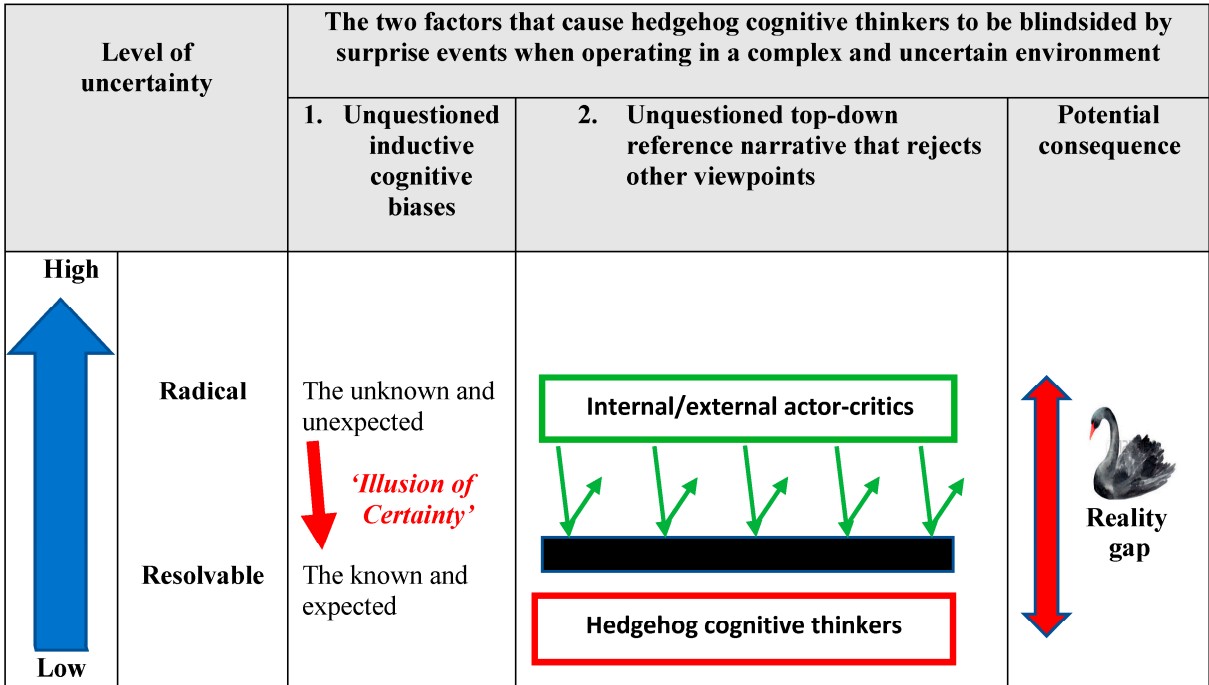

**Figure 1.** The two factors that cause hedgehog cognitive thinkers to be blind to 'black swan' events.

### 2.6. The Simplified Cynefin Sense-Making Framework

An initial review considered different uncertainty models, such as the Stacey matrix [22], the Ashby Space diagram [23], the probable/preferable/possible/plausible framework and the aleatory/epistemic framework. This research used the simplified Cynefin sense-making framework [24] with its 'ordered'–'unordered' sets of domains as a useful additional lens for viewing and supporting the description of Knightian risk/uncertainty.

Kurtz and Snowden [25] state that the Cynefin framework was developed to reflect and describe the evolutionary nature of complex systems and their inherent uncertainty. The framework sorts the issues into five domains defined by the nature of the relationship between perceived cause and effect. Four of these relationships, being simple/obvious, complicated, complex, and chaotic, require decision-makers to diagnose situations and to act in contextually appropriate ways. The fifth domain disorder is the state of not knowing what type of causality exists, and is where people will revert to their own comfort zone

in making a decision. No domain is more desirable than any other as it is not a value-based system. The framework is used to consider the dynamics of situations, decisions, perspectives, conflicts, and changes in order to come to a consensus for decision-making under uncertainty [24,25].

In Figure 2 below, the two right-side domains reflect a composite 'order' domain of what is known and what is knowable. This contrasts with the left-side domain of unorder, where distinctions of knowability are or may be less important than distinctions of interaction—that is, distinctions between what we can 'pattern' (complexity) and what we need to stabilise in order for patterns to emerge (chaotic) [25]. What is of concern for this research is that in most organisations, a state of order, or at least a perception of order, seems to dominate and only a real crisis will change this [26].

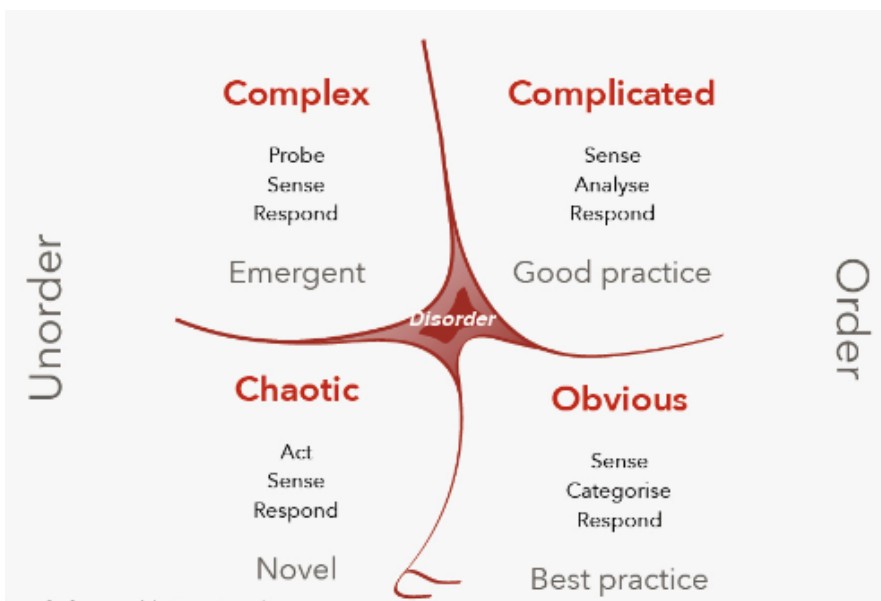

**Figure 2.** The basic Cynefin framework. Source: Kurtz and Snowden [25], with updated headings from Snowden [27].

### 2.7. Complex Systems

Over time, most social systems have become complex with tight coupling (often through social media). Coupling is tightened with very tight schedules, new systems and no margin for error [28]. As systems add webs of interactions, relationships and interdependencies, complexity increases and positive/negative feedback loops emerge, creating non-linear effects [29]. Complex systems are varied and include future markets, entrepreneurial business, nations, stock markets, and international politics. Past a certain point, the internal dynamics of these systems is bewilderingly unknowable, with unsuspected risk. Most situations and decisions in organisations are complex because some major change such as a bad quarter, a shift in management, or a merger or acquisition, introduces unpredictability and flux [24]. Complex systems are emergent in nature, arising from the interactions of many agents, which are dynamic, so reliance on historical trends will not prepare decision-makers for the new unexpected patterns [25,30].

### 2.8. Knightian Uncertainty/Risk Requires Two Different Sets of Assumptions and Techniques

Understanding which Cynefin domain that an issue in question fits into is extremely important. While all four of the quadrants of the Cynefin framework are different, this study focusses on Cynefin's important 'unordered' versus 'ordered' domain 'fault' line, since that separates the 'complex' versus 'complicated' domain divide which is where most strategic thinking/planning is situated. (See Table 1). The two approaches are radically different and they use completely different thinking approaches, methods, and tools. For

example, whereas wicked problems clearly fit into the complex domain, tame problems fit into the complicated domain [31,32].

**Table 1.** The different management practices/approaches that need to be used for Knightian risk and Knightian uncertainty.

| Subject | Knightian Risk—Resolvable Uncertainty | Knightian Uncertainty—Radical Uncertainty | Source |
|---|---|---|---|
| Cynefin domains | 'Ordered' domains<br>- 'Obvious' (i.e., knowns)<br>- 'Complicated' (i.e., knowables) | 'Unordered' domains<br>- 'Complex' (i.e., unknowns)<br>- 'Chaotic' (i.e., unknowables) | Kurtz and Snowden [25], Snowden and Boone [24], McLeod and Childs [33] |
| Cause/effect | - Clear cause and effect but sometimes separated over time and space<br>- One or more right answers | - No cause-and-effect relationship or only in retrospect<br>- No right answer, but emergent patterns | Kurtz and Snowden [25], Snowden and Boone [24] |
| Techniques | - Fact-based management<br>- Predict, plan and control<br>- Analytical—reductionist<br>- 'If then' deductive logic | - Pattern-based leadership<br>- Explore, experiment, and learn<br>- Perspectives<br>- 'What if' counterfactuals | Kurtz and Snowden [25], Snowden and Boone [24], Tetlock and Gardner [34] |

Following are examples of this Knightian divide of resolvable versus radical uncertainty, from different aspects of management. For example: deliberative versus emergent strategies [35]; tame problems versus wicked problems [36]; detailed complexity versus dynamic complexity [37]; forecasting versus foresight [38]; and 'normal' incremental science versus transformational paradigm shifts [39].

## 3. Methods

### 3.1. Selection of a 'Black Swan' Case Study

The 2010 Pike River coal mine explosion [2] was selected as the case study from a list of New Zealand Government-initiated public inquiries over the last thirty years, using three criteria: recency (for availability of records), significance of the consequence (i.e., loss of life/long-term effect) and complexity of the situation (i.e., multiple stakeholders/layers with different perspectives). These criteria ensured that the case study had good documentation (via a subsequent Royal Commission of Inquiry) on the different viewpoints and interpretations of events ex-ante, especially those that disagreed with Pike's official reference narrative and thus those who held them were not necessarily as surprised as the key decision-makers when the mine 'unexpectedly' exploded.

### 3.2. Key Facts and Timeline

Pike River Coal Ltd. (Pike) was publicly listed in 2007 (refer to Table 2), with the aim of developing a coal mine at Pike River, near Greymouth in the South Island of New Zealand. The mine was located in a very difficult and remote operating environment. Not only were the geography and geology difficult, including the coal seam saddling an earthquake fault line (see Figure 3), but it was a high rainfall area and it snowed in the winter. Despite this, Pike set very ambitious targets and considered itself a 'showpiece' for modern coal mining. Pike consequently experienced unexpected difficulties, which led to significant cost and time overruns. The mine was still being developed at the time of the methane explosion in 2010. In the three years after Pike's initial public offer (IPO) in 2007 that raised $85 m, Pike had gone to shareholders three further times for an extra $155 m and at the time of the explosion were seeking from them another $70 m. The explosion killed 29 miners and ended all mining at the site. All funds invested (+$300 m) were lost [40].

**Table 2.** Timeline of key events leading up to the methane explosion in the Pike River coal mine.

| Dates | Key Events |
| --- | --- |
| 20 July 2007 | Pike listed as a public company. |
| 17 October 2008 | Date of first coal. |
| 27 November 2008 | Formal mine opening. |
| February 2009 | Ventilation shaft collapses. |
| 19 February 2010 | First coal export shipment. Graben (rockfall in path) identified and it took months to penetrate. |
| 5 July 2010 | Board proposes a bonus scheme to get the mine ready for production. |
| 6 September 2010 | Second coal export shipment. |
| 10 September 2010 | Pike Board dismisses Gordon Ward as CEO and replaces him with Peter Whittall. |
| 19 September 2010 | Start of hydro-mining. |
| 19 November 2010 | Mine explosion. |

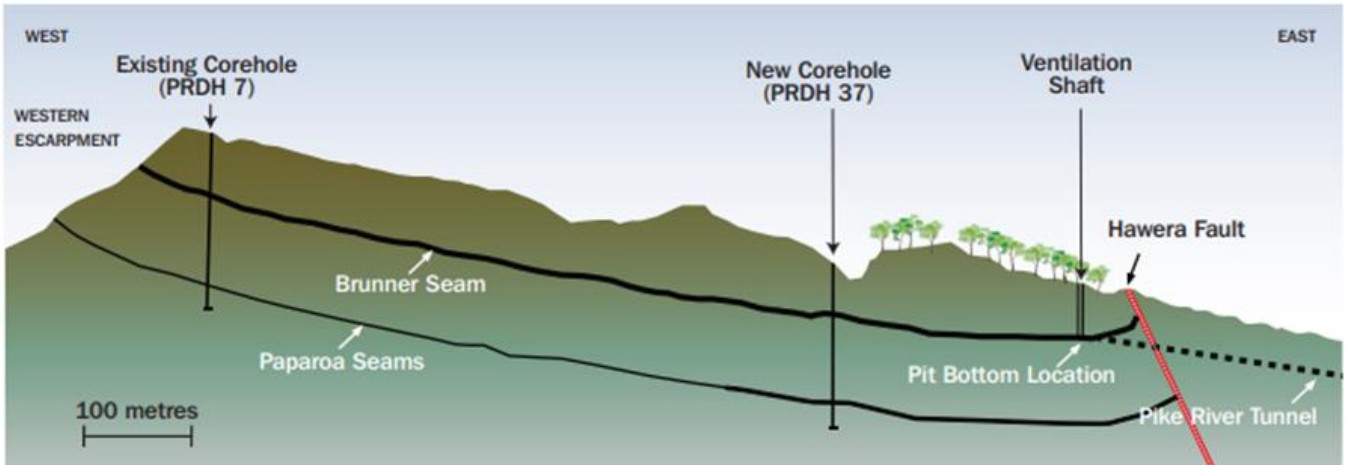

**Figure 3.** Pike River mine cross section showing the two seams of interest. Source: Pike [41].

### 3.3. Pike's Management Experienced a Sequence of Significant Unexpected Events

The Pike mine was still being developed despite serious gaps or issues being identified in all major mining systems, including gas monitoring, methane drainage, the ventilation system, the underground electrical system and the hydro-mining system [4].

As Pike operated in a highly uncertain and complex environment, which many of the actor-critics tried unsuccessfully to get acknowledged, significant unexpected (at least for the three decision-makers) events kept occurring. These included; the 'unexpected' poor quality rock in building the 2.3 km entrance tunnel (September 2006 to October 2008); the 'unexpected' shaft collapse when building the 111-m ventilation shaft (mid 2008 to June 2009); and the unexpected 200-metre 'graben' of shattered rock in June 2009 where there was meant to be coal. Unfortunately, Pike's three key decision-makers did not seem to learn from these unexpected events, since there existed among them a commonality of issues of poor geological knowledge, significant understatement of costs, significant loss of time, and apportioning blame to others [40].

### 3.4. Research Approach

A strong feature of this research was the wide cross-disciplinary nature of the different elements relating to 'uncertainty'. For example, bounded rationality and cognitive biases (i.e., behavioural psychology), 'black swan' events (i.e., risk management), 'the problem of

induction' (i.e., philosophy), 'normal accidents' (i.e., complexity), Knightian risk and uncertainty and narrative economics (i.e., economics), fox/hedgehog cognitive thinking styles (i.e., political science/forecasting), the Cynefin framework (i.e., management), and 'drift charts' and 'Swiss Cheese' models (i.e., accident investigation). The research started with a wide review of all these interconnected literatures. The research involved a qualitative exploration and understanding of the multiple layers and dimensions within the process of decision-making under uncertainty in a large complex organisation. Multiple conceptual models and case study voices were considered. This approach fitted a constructivist approach or constructivist interpretative paradigm [42].

The use of a 'narrative' as a frame, framework or construct is common in the social sciences or in futures research [43]. However, in economics, using narratives is a relatively novel concept, since mainstream economics has historically assumed that preferences are located exclusively with the individual, and economic models deal exclusively with decision-making under Knightian risk (i.e., known or assumed probabilities) rather than under Knightian uncertainty (i.e., unknown probabilities) [21]. More recent approaches to economics such as feedback economics [44] or narrative economics [10,45] challenge that approach.

### 3.5. Lens for Analysis—Defenders Versus Challengers of Pike's Reference Narrative

The study needed to determine Pike's reference narrative so that those challenging it ex-ante could be identified, and to clearly distinguish the defenders from the challengers. The unit of analysis applied was the three key decision-makers: the Board Chair, the CEO (from 2007 to September 2010, departing 6 weeks before the explosion) and the General Manager (Mines) 2005–2010, then CEO from October 2010, supported by the Pike Board and some senior managers [4,40,46]. The views of the three key decision-makers represented the reference narrative of Pike, as the three key decision-makers signed off all public disclosures ex-ante. The Royal Commission of Inquiry brought to light ex-ante information not previously publicly available about the thinking of the three key decision-makers and of a number of internal/external actor-critics. For all issues, the views were split between the three key decision-makers who were active defenders of Pike's reference narrative versus the actor-critics who challenged all or part of that narrative. The focus was exclusively on the reaction of the three key decision-makers to the ex-ante and ex-post criticism. This split in views between defenders and challengers of Pike's reference narrative was highlighted by the analysis techniques used during the research. These included the drift over time charts [29], forced scenario process [47–49], and reading the transcripts of the Royal Commission hearings and the Royal Commission's final report [4,46]. The research was undertaken with full awareness of the inherent problems of analysing past events, especially the need to address hindsight and outcome biases, both in doing this research and in assessing what actors were saying ex-post about what happened ex-ante.

### 3.6. Synthesis and Bricoleur Process—Making a Pattern of All the Findings

Using a constructivist interpretation paradigm requires the researcher to be responsible for the credibility of the research, as traditional positivist notions of validity and reliability are never captured fully. Rather, it is the researcher as a bricoleur [50] who makes a series of interpretations based on a number of different models and approaches in order to construct an emergent conceptual pattern with new insights. The bricoleur does this by synthesizing all the individual parts into a complex quiltlike pattern. The process depends upon capturing a wide variety of knowledge, then interpreting this in a meaningful way, which can be explained and justified using notions of credibility, transferability, dependability, and confirmability. This is the skill of the bricoleur or quilt maker [50,51]. This type of research approach is difficult to describe in advance because of uncertainties about methods needed and results that will emerge. Non-linear and non-sequential research will be messy, especially in comparison to linear and sequential research, but the researcher

needs to accept and be comfortable with that reality. Figure 4 provides a simplified summary of the researcher's non-linear bricoleur journey.

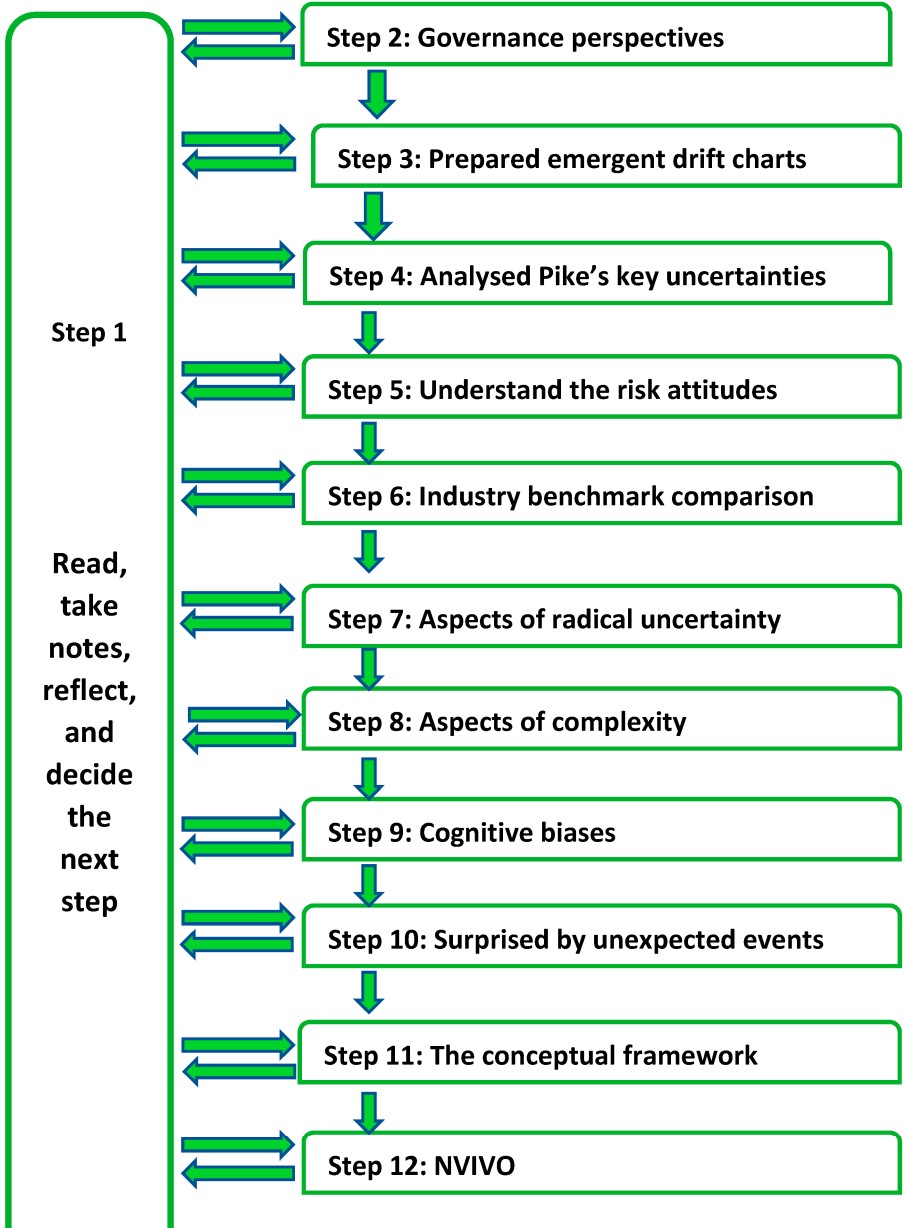

**Figure 4.** Summary of the bricoleur process.

## 4. Results

### 4.1. 'What' Pike's Key Decision-Makers Thought—Pike's Unquestioned Top down 'Reference Narrative'

From 2007 when it became publicly listed, Pike promised that once its mine was developed, it would produce 1m tonnes per year for 18 years (i.e., 17.6 m tonnes) of premium, hard coking, ultra-low ash coal (i.e., the lowest in the world), being highly priced, and easy and safe to extract [52]. These ambitious aspirations were repeated in multiple shareholder publications [41,52–54] up to the 2010 methane explosion. This aspiration formed a complex multifactor reference narrative.

All parts of the reference narrative needed to be realistic otherwise there was the threat of some type of failure, i.e., production, financial, or safety. In reality, Pike's reference narrative was not adjusted over time despite conflicting information emerging. Pike's

public statements of progress and achievement were often far rosier than they should have been, considering the significant time delays of over three years, cost overruns and the small amount of coal Pike had actually shipped (i.e., 0.044 m tonnes) by the time of the explosion [54,55]. The coal produced failed to meet the promised quality specifications, being around 5% ash compared to the promised ultra-low ash of 1%. From the subsequent information raised during the Royal Commission hearings, there was a question mark over every component of Pike's reference narrative and red flags for financial viability, safety, coal reserves, coal production estimates and coal quality.

In hindsight, part of the reference narrative is what Akerlof and Shiller [56] (p. 55) call 'new era' narratives, since they purport to describe historic changes that will propel the company into a brand-new era. New era narratives have tended to accompany the major booms in stock/commodity markets around the world. Pike's 'new-era' narrative stressed that Pike was a publicly listed start-up company, that promised to be an industry leader that would set new standards of excellence in thinking, systems, equipment, and productivity. The comparison to Pike's 'showcase' profile was the state owned Solid Energy Ltd., with its highly unionised workforce, average productivity and its hidebound traditions [40]. Pike's public listing coincided with a surge in the international demand for coking coal from China. Pike's reference narrative existed within this coal price 'boom', which meant Pike believed at the peak of the boom that their potential coal reserves were worth between $2.3 b to $4 b [55]. By 2012, this international coal demand from China had collapsed, as had world coal prices, immediately making most coal mining, especially underground mining in New Zealand, uneconomic.

*4.2. 'What' Stakeholders Thought of Pike's Reference Narrative Determined the Level of Surprise to the 2010 Pike Mine Explosion*

The explosion was a 'black swan' event to many people, but there were four different levels of surprise to it. The first group taken by complete surprise was the three key decision-makers and their supporting in-group (i.e., the most hedgehog-like cognitive thinkers), who formulated, implemented, and defended Pike's reference narrative. They believed Pike had good systems of control that would block any black swan event. The second group was also apparently completely surprised. This group included mostly passive investors, many or most staff, most politicians, and the public. This group relied on the optimistic public statements from the three key decision-makers for their information. Since Pike had always said that they had good systems of control and in the absence of any other information, this was what this group believed. Numerically, this is the largest of the four groups. A third group that was surprised (but not completely surprised!) by the 'black swan' event consisted of managers, staff, and contractors who were aware of specific problems, which had troubled them enough to have (in the event, unsuccessfully) raised their concerns with higher management. Included in this group were: Doug White (General Manager 2010) who regretted not having a tube-bundle gas monitoring system or second egress; Neville Rockhouse (Safety and Training Manager) who regretted Pike had only an unusable shaft as a second exit; and Pieter Van Rooyen (Technical Services Manager 2009–2010) who regretted the lack of a comprehensive mine master plan that linked gas drainage, geology, etc. While all these people would have been surprised by the explosion, they all saw different, but equally important, aspects of Pike's operations that deeply concerned them, which, if addressed, could have prevented the explosion. Each of these examples is a strong hint that the system was at risk of being in trouble. The fourth group consisted of the few internal/external actor-critics (i.e., the most fox-like cognitive thinkers), who were often frustrated or marginalised by decisions, and were deeply troubled with a bad feeling about Pike in a holistic sense. This group understood that the parts that concerned them were linked to other parts that would affect the whole system, and therefore they expected or were concerned that a system failure would occur. People in this group did not consider the Pike mine explosion a 'black-swan event'. These four groups are summarised in Table 3.

**Table 3.** 'What' stakeholders thought of Pike's reference narrative determined the level of surprise to the 2010 Pike mine explosion.

| | Attitude to Pike's Reference Narrative | Category of Stakeholder | Cognitive Thinking Style | Level of Surprise |
|---|---|---|---|---|
| 1 | Actively defended. | The three key decision-makers & their supporting 'in-group'. | The most hedgehog-like. | A complete surprise. |
| 2 | Relatively passive acceptance. | Passive stakeholders (majority). | Moderate hedgehog-like. | A complete surprise. |
| 3 | Troubled by specific aspects. | Informed, mostly active internal stakeholders. | Moderate fox-like. | A surprise, but were aware of problems. |
| 4 | Believed it was seriously flawed. | Informed, mostly 'out-group'. | The most fox-like. | A surprise, but were aware of deep problems. |

*4.3. 'How' Pike's Key Decision-Makers Thought—Pike Senior Management's Overconfidence Was Unquestioned*

In the three years leading up to the mine explosion, Pike's three key decision-makers showed evidence of various inductive cognitive biases, such as the optimism bias, overconfidence and the resulting planning fallacy bias (i.e., overpromise and underdeliver). Over time, this led on to the 'sunk cost' fallacy and to actions in the fourth quadrant of prospect theory [57], where they promised that 'hydro-mining' would dramatically improve production levels, despite all the evidence to the contrary.

Pike's CEO (an accountant by profession) consistently used ambitious 'stretch' targets for practically all indicators all of the time. (Note, it is common for CEOs to be extremely confident with a clear narrative/vision [58]). This contrasted with the actor-critics who consistently wanted more modest 'fit objectives', being ones that could be achieved in the difficult circumstances that Pike was operating under. Under cross examination, the Board Chair said: *'We did have issues with overpromising and under-delivering'* [46]—December 2011 (p. 3931). *'Now my understanding was that Mr. Ward was being provided with a wider range of alternative outcomes but was electing at the more optimistic end of performance... to keep assuming or factoring in performance at the more optimistic end of the likely range of outcomes'* [46]—December 2011 (p. 3959).

Groupthink kept the small in-group together (and kept them blind); and the conformity bias, organisational silence and obedience kept the much larger out-group quiet. This culture of silence and lack of power was exacerbated by having a high number of inexperienced workers, a high number of foreign workers and high number of contractors. There was a high turnover of staff and middle management throughout the entire period, which the Board Chair rationalised was due to the mining boom occurring at the time [4].

## 5. Discussion and Conclusions

The 2010 mine explosion was completely unexpected by Pike's key decision-makers. It had tragic consequences for the 29 who died, and for all those involved. It was an economic blow for Pike's shareholders, the local community and for New Zealand as a whole, and many wondered how this accident could occur in 2010 (i.e., modern times).

With the benefit of hindsight and a Royal Commission of Inquiry, we can determine that Pike's three key decision-makers (with their supporting Pike/NZO&G Boards) displayed what Tetlock [3] calls strong 'hedgehog' cognitive thinking, in two different ways. The first way involved unquestioned inductive cognitive biases (affecting 'how' they thought), such as the optimism bias, the planning fallacy, the sunk cost fallacy, and confirmation bias [19,57]. A second and related aspect of 'hedgehog' cognitive thinking Pike's three key decision-makers exhibited was their confidence in their collective top-down reference narrative (affecting 'what' they thought), which was neither robust nor resilient.

Unfortunately, they actively defended this narrative from internal or external challenge. Overall, the unquestioned inductive biases and the unquestioned reference narrative meant that there were unrecognised uncertainties, since both simplifications focus on the known and certain, without understanding that by doing so, they overlooked or underappreciated uncertainty and risk.

There was no one on the board or senior management who publicly (ex-ante or ex-post) demonstrated fox-like cognitive thinking, and/or the necessary organisational authority to successfully challenge and overturn the collective thinking of the top team. This limited Pike's collective intelligence to that of a small number of people at a time when the collective intelligence could have been expanded to include, for example, a range of staff and consultants. Their 'hedgehog' cognitive thinking contributed to Pike's three decision-makers becoming 'wilfully blind' [59] in both 'what' they thought and 'how' they thought. This is summarised below in Figure 5.

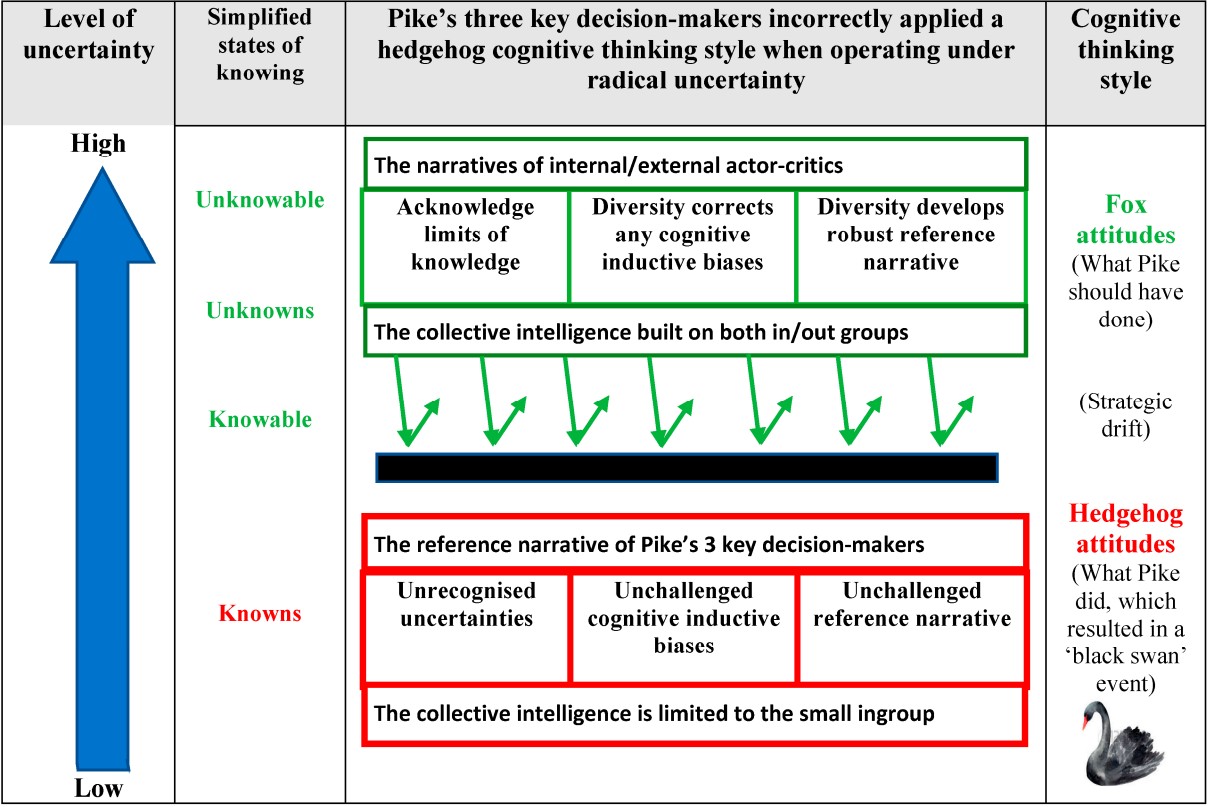

**Figure 5.** Pike's three key decision-makers were blinded by hedgehog cognitive blindnesses.

### 5.1. Limitations and Future Research Directions

The limitations of this study are in line with the general limitations applied to a single exploratory qualitative case study, in just one country (in this case, New Zealand), which has the potential for limited generalisability. However, case studies can provide rich insights especially when it comes to a 'revelatory case' [60], and in this research, the issue is not perceived as a problem. The research can be used as a means of assessing the utility and transferability of the two common types of organisational blindness to other contexts and other decision-makers, in both ex-ante and ex-post settings.

Another general limitation inherent from involving any form of 'narrative', as well as all forms of business and economic decision-making 'models', is that these only represent parts of the target system, by simplifying key aspects, which is never as complete as the actual system itself [43]. A third limitation (but also a great strength) is that this research is multidisciplinary and that can run against the grain of modern academia, even within the business school [61], which rewards deep specialization.

*5.2. Theoretical Contribution*

This research builds on the Knightian uncertainty and 'uncertainty aversion' literature, by identifying two common but overlooked organisational oversimplifications that indicate potential 'uncertainty aversion' when operating under radical uncertainty. The two interconnected oversimplifications affected both 'how' and 'what' the key decision-makers thought.

*5.3. Practical Contribution*

This research provided an additional lens to review the organisational tragedy at Pike River, which resulted in a different conclusion from what the Royal Commission of Inquiry concluded at the time using the 'Swiss Cheese' model. The Inquiry concluded that management had put 'production over safety', which management consistently denied. This research found the three key decision-makers were wilfully blind due to at least two common and interconnected organisational cognitive simplifications of uncertainty.

This result is also different from other studies of the Pike Mine disaster that focused purely on issues of: health and safety; human resources; or systems approaches to the mine operations (for example, [62,63]). Those studies had different research questions, different lenses and methodologies and arrived at different conclusions, appropriate for the relevant discipline. In a complex situation such as Pike River, there is a need for all the different approaches, to ensure that all the key elements are considered holistically, rather than in different discipline silos.

The research has a much wider practical contribution than considering the decision-making at Pike River, in as much as it provides an additional set of lenses for both ex-post and ex-ante analysis of business and public policy decision-making under uncertainty, including reviews of decision-making failures. This would be useful for political/commercial managers, company directors, and accident investigators. For example:

- Boards/management should include both fox and hedgehog cognitive thinkers, when there is the possibility of great uncertainty/complexity. Both types of thinkers can add value, but since boards need, among other things, a future focus, with awareness of high-cost, low-probability events, having one or more fox-like cognitive thinkers is critical.
- The research highlights the importance of ex-ante decision-making under high uncertainty/complexity, and the need to amend the reference narrative as required. In this type of situation, the decision-makers should be doing more than monitoring their reference narrative. New information and new interpretations of that information will mean that the decision-makers will need to actively revise their reference narrative. At some stage, if the complexity/uncertainty is high, then decision-makers must be prepared for a paradigm shift in thinking and a complete revision of their reference narrative.

The package of lenses used in this research provide a different and additional set of lenses for reviews that could be applied to past or future Commissions of Inquiry that involve unexpected disasters and decision-making failures. This is especially so if there is evidence available to the Inquiry of a strong actor-critic involvement ex-ante, to establish whether strategic drift had occurred and the relevant board/management had a relatively static reference narrative, and thus whether these hedgehog cognitive blindnesses caused key decision-makers to be surprised by a 'black swan' event.

**Author Contributions:** Conceptualization, R.J.L.: methodology, R.J.L.; investigation, R.J.L.; writing—original draft preparation, R.J.L.; writing—review and editing, R.Y.C., B.E.H. and I.Y.; supervision, R.Y.C., B.E.H. and I.Y. All authors have read and agreed to the published version of the manuscript.

**Funding:** This research received no external funding.

**Data Availability Statement:** Data are contained within the article.

**Conflicts of Interest:** The authors declare no conflict of interest.

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
