# Peer review of "Why Do Key Decision-Makers Fail to Foresee Extreme ‘Black Swan’ Events? A Case Study of the Pike River Mine Disaster, New Zealand"

_systems, doi:10.3390/systems12010034_

Round 1
Reviewer 1 Report
Comments and Suggestions for Authors
The reviewed paper addresses the strategic issue of why key decision-makers fail to predict potential extreme ‘black swan’ events. However, it is poorly explained why this problem was chosen, at least in the abstract and introduction.
There are also some technical issues that need to be corrected (e.g fig 4 is of poor quality, fig 5 - some captions are too cropped).
In general, the paper is interesting and fits into the thematic scope of the Journal's section 'The Systems Thinking Approach to Strategic Management'.
Reviewer 2 Report
Comments and Suggestions for Authors
The article titled "Why Do Key Decision-Makers Fail to Foresee Extreme 'Black Swan' Events? A Case Study of the Pike River Mine Disaster New Zealand" presents a comprehensive analysis of why key decision-makers often fail to anticipate or react appropriately to unexpected, extreme events, using the Pike River Mine disaster as a case study.
Interdisciplinary Approach is a kind of strength of the article (although the authors declared it as a limitation as well). The Pike River Mine disaster is analyzed in detail, providing a thorough and nuanced exploration of the event. This comprehensive approach helps to validate the theoretical constructs discussed. However, the article also has some drawbacks. One of them is the methodological part. Despite providing three conditions for selecting a case study (recency, significance of the consequence, complexity of the situation), it is not entirely clear why this particular case - i.e., the other cases taken into account did not meet these conditions? Generalization in the single case study method is difficult, and if the case is not "typical" in any way, what general conclusions can be drawn. Moreover, sentences like: "The research started with a wide review of all these interconnected literatures. The research involved a qualitative exploration and understanding of the multiple layers and dimensions within the process of decision-making under uncertainty in a large complex organization. Multiple conceptual models and case study voices were considered", sound mysterious (they give the impression of science and reliability, but due to the lack of specifics they contribute nothing).
Another point of discussion is a clear definition of how the four groups indicated in lines 351-378 were distinguished. We have the characteristics of the groups and who belongs to them (key decision-makers, passive investors, most staff, etc.). Have these groups been studied in any way – how was this determined? Moreover, statements such as: There was no one on the board or senior management who demonstrated fox-like cognitive thinking…..” should also be supported by facts (are you sure no one? How do you know?). Besides it should be remembered, the reliance on secondary data and the retrospective analysis of the disaster might introduce biases, particularly hindsight bias, which could influence the interpretation of events and decisions. Despite a separate “Practical Contribution” section, the contribution to practice is shallow and vague. This part should be strengthened with more specific elements.
Comments on the Quality of English LanguageNo comments
Reviewer 3 Report
Comments and Suggestions for Authors
This article utilizes secondary data sources from the Royal Commission of Inquiry regarding the Pike River Mine Disaster in New Zealand and other relevant literature. It employs a case study method to analyze why key decision-makers fail in the face of extreme, unpredictable, and impactful black swan events. The study aims to reveal the challenges of making effective decisions in highly uncertain environments but may have the following issues:
1. The Pike River Mine Disaster in New Zealand, as a classic case, has been studied by many scholars from various aspects, including from the perspectives of managers, management systems, and communication mechanisms. This article chooses these two angles related to managerial blindness. During the review process, I observed that Reference 2 and this article share similar research angles, content, and some methodologies. Please elaborate on how this article differs or advances from the research conducted in Reference 2 and other scholars in relevant areas.
2. The Simplified Cynefin Sense-Making Framework used by the author is a simplification of the traditional Cynefin Sense-Making Framework. However, the author did not specify why a simplification was necessary. Additionally, the distinction between 'complex' and 'complicated' is minimal, which may not clearly delineate the difference for readers.
3. Narrative Economics, as an important research foundation of this article, lacks an introduction to similar studies using this method.
6. The author conducted a Bricoleur process, a comprehensive analysis process using qualitative methods. Please elucidate the scope and reliability of this research method. For instance, detail the selection of research design, data collection, and analysis process, and clarify whether changes in the information/data used in the research process or due to the researcher's access to limited information could impact the research results.
7. Relevance of Section 4.1 as Research Results: Please review whether it is necessary to include section 4.1 as part of the research results, as it represents an actual occurrence rather than your research findings. It is suggested to be incorporated into section 3.2.
8. While the article provides rich insights through an in-depth exploration of the Pike River Mine disaster case, it should compare the conclusions derived from this study with previous investigation conclusions or research on this event, thereby highlighting the actual contributions of this research.
Some layout issues:
1. Title Appropriateness in Section 2.1: The main content of section 2.1 broadly includes two aspects. The first part explains Knightian Risk and Uncertainty, while the latter half discusses various scholars' debates on the concept of uncertainty. It is evident that section 2.1 primarily involves two different concepts/views, rather than methods.
2. Numeration Error in '3.3 Lens For Analysis – Defenders Versus Challengers of Pike’s Reference Narrative': It should possibly be numbered as 3.4.
3. Layout Issues in Figure 5: Please check if there are any layout issues with Figure 5.
4. Extraneous Punctuation in Title 4.3: Please review if there's any unnecessary punctuation in the title of section 4.3.3.
Reviewer 4 Report
Comments and Suggestions for Authors
Dear authors, thanks for submission of your paper. The paper covers an interesting issue of why key decision-makers fail to foresee potential extreme ‘black swan’ event.
The topic is original and results suggest approaches to avoid mistakes in predictions and expand the forecast.
The manuscript is well-prepared and organized. The methodology part is explained in detail and supported by references. Authors can add a brief review of similar studies to show the development in this field.
The structure and methodology of the paper are quite unique and ,therefore, it expands the current literature on this topic.
Discussion and Conclusions are supported by the results presented in the manuscript.
The conclusions are built based on the obtained results of the study and they address the main question of the research.
The paper has significant theoretical and practical contributions, which are discussed and explained in detail.
References used in the paper are relevant and up-to-date. The tables and figures are well explained in the text and they enrich the content of the paper.
Based on the above-mentioned, the paper can be recommended for publication in its present form.
Round 2
Reviewer 2 Report
Comments and Suggestions for Authors
Most of the previous comments have been taken into account and corrected
Comments on the Quality of English Languageno comments
Reviewer 3 Report
Comments and Suggestions for Authors
1. In response to the author's reply to the first comment, if, as the author states, the work is merely a development and simplification, while this does not negate its progressive value, it may imply that the originality, theoretical, and practical contributions of this article could be somewhat limited.
2. We understand what the author intends to convey in response to comment 5, yet we continue to hold our original view on this matter.
3. Regarding comment 6, our intention is to advise the author to enhance the paper by adding a section that compares this research with earlier studies that have investigated similar themes. This is intended to more effectively highlight the theoretical and practical advancements of the study, rather than just contrasting it with the Royal Commission.
Round 3
Reviewer 3 Report
Comments and Suggestions for Authors
The reviewer is satisfied with the revisions made to the manuscript and has no further issues; the paper can be considered for acceptance.